# *KEAP1*-Mutant Lung Cancers Weaken Anti-Tumor Immunity and Promote an M2-like Macrophage Phenotype

**DOI:** 10.3390/ijms25063510

**Published:** 2024-03-20

**Authors:** Christopher J. Occhiuto, Karen T. Liby

**Affiliations:** 1Department of Pharmacology and Toxicology, Michigan State University, East Lansing, MI 48824, USA; occhiut1@msu.edu; 2Department of Medicine, Indiana University School of Medicine, Indianapolis, IN 46202, USA; 3Department of Pharmacology and Toxicology, Indiana University School of Medicine, Indianapolis, IN 46202, USA

**Keywords:** NRF2 pathway, KEAP1, lung cancer, immunosuppression, macrophages, M2 polarization

## Abstract

Considerable advances have been made in lung cancer therapies, but there is still an unmet clinical need to improve survival for lung cancer patients. Immunotherapies have improved survival, although only 20–30% of patients respond to these treatments. Interestingly, cancers with mutations in Kelch-like ECH-associated protein 1 (*KEAP1*), the negative regulator of the nuclear factor erythroid 2-related factor 2 (NRF2) transcription factor, are resistant to immune checkpoint inhibition and correlate with decreased lymphoid cell infiltration. NRF2 is known for promoting an anti-inflammatory phenotype when activated in immune cells, but the study of NRF2 activation in cancer cells has not been adequately assessed. The objective of this study was to determine how lung cancer cells with constitutive NRF2 activity interact with the immune microenvironment to promote cancer progression. To assess, we generated CRISPR-edited mouse lung cancer cell lines by knocking out the *KEAP1* or *NFE2L2* genes and utilized a publicly available single-cell dataset through the Gene Expression Omnibus to investigate tumor/immune cell interactions. We show here that *KEAP1*-mutant cancers promote immunosuppression of the tumor microenvironment. Our data suggest *KEAP1* deletion is sufficient to alter the secretion of cytokines, increase expression of immune checkpoint markers on cancer cells, and alter recruitment and differential polarization of immunosuppressive macrophages that ultimately lead to T-cell suppression.

## 1. Introduction

Advances in lung cancer research have led to better patient outcomes for a disease with historical 5-year survival rates of only 22% [1,2]. In the last decade, median survival doubled from 8–17 months with chemotherapy alone to 18–36 months with targeted treatments and immunotherapy [3]. While survival has significantly improved, only 20–30% of patients with non-small cell lung cancer (NSCLC) respond to immunotherapies [4]. The development of new targeted therapies and a more comprehensive understanding of tumor-immune system biology will help address the clinical need to improve outcomes.

Nuclear factor erythroid 2-related factor 2 (NRF2) is a transcription factor that regulates antioxidant and cytoprotective genes [5]. The NRF2 protein is constitutively expressed but under constant suppression by its negative regulator, Kelch-like ECH-associated protein 1 (KEAP1), which targets NRF2 for proteasomal degradation [6]. Consequently, KEAP1 inactivation leads to constitutive NRF2 activity. An array of cytoprotective genes is increased by NRF2 activation, including *NQO1*, *HMOX1*, *GCLC*, *GCLM*, *SOD1*, *ABCB1*, and *GST* genes [7]. These genes are known to control cellular redox responses and detoxification. Other pathways regulated by NRF2 involve processes such as glycolytic regulation [8], pentose phosphate pathway regulation [9], amino acid metabolism [10], autophagy [11], and mitochondrial biogenesis [12]. NRF2 has many known roles as a transcriptional regulator in detoxification and rewiring of cell metabolism, but new discoveries are expanding knowledge of its regulation of inflammation.

Studies over the past decade showed that NRF2 promotes an anti-inflammatory phenotype in immune cells [13]. The impact of immune cell-specific NRF2 activation includes blocking proinflammatory cytokine transcription in M1-like macrophages [14,15]; promoting an M2-like macrophage secretory phenotype [16]; decreasing inflammatory cytokine transcription in lymphocytes [17]; and downregulating the NF-kB pathway [18,19]. Despite these intriguing findings in the immune system, there is a paucity of studies relating NRF2 activation and anti-inflammatory activity in cancer cells, which is important since cancers with *KEAP1* mutations are resistant to immune checkpoint inhibition [20]. This topic is surprisingly understudied, given that NRF2-activating mutations are found in up to 30% of NSCLCs, with *KEAP1* mutated in nearly 20% of lung adenocarcinomas [21]. *KRAS*, a critical oncogenic driver of lung cancer, is frequently co-mutated in lung cancers: 27% of patients with *KRAS* mutations also have deleterious mutations in *KEAP1*/*NFE2L2* genes, comprised of 24% *KEAP1* and 3% *NFE2L2* co-mutations [22]. The frequency of co-mutations of *KEAP1* and *KRAS* in lung cancers, which further increases treatment resistance [23] and decreases survival [22], adds to the importance of addressing the biologic complexity and immune regulation by these tumors. 

Cancer cells express and secrete a wide variety of immunomodulatory mediators [24,25]. Immune regulation is critical for cancer survival because the tumor immune microenvironment (TIME) can both promote tumor growth and kill cancer cells [26,27,28]. Data from The Cancer Genome Atlas suggest lung adenomas with high *NQO1* (a downstream target of NRF2 and surrogate for NRF2 activation) mRNA expression exhibit immunosuppressive phenotypes with lower representation of genes relating to immune subsets [29,30]. Indeed, CD8^+^ T cells, NK cells, and B cells are decreased in *KEAP1*-mutant murine lung tumors with constitutive NRF2 activity [20,29]. There is a disconnect in how NRF2 activity in lung cancer cells informs the immune competency of the tumor microenvironment and promotes lung tumor progression. Although correlative, these studies suggest immune regulation by NRF2-activated cancers, and understanding specific mechanisms would be beneficial in tailoring targeted treatments for cancer patients. 

A recent publication by Zavitsanou et al. [31] investigated the role of *KEAP1*-mutant lung cancers in promoting immunosuppression. *KEAP1*-mutant lung cancers decreased the recruitment of CD103^+^ dendritic cells and therein lowered CD8^+^ anti-tumor immunity. Zavitsanou et al. generated a large single-cell sequencing dataset that we have utilized in our studies to complement the investigation of immune microenvironmental changes mediated by *KEAP1* cancer mutations. The studies described here evaluate the immune regulating capability of *KEAP1*-mutant cancer cells and how they further promote immunosuppression through the polarization of macrophages. 

## 2. Results

### 2.1. Development and Validation of KEAP1 and NRF2 Knockout Cancer Cell Lines

We sought to characterize immune microenvironmental alterations caused by cancer cell-intrinsic *KEAP1* mutations while preserving the NRF2 status of immune cells. To achieve this, we selected a murine Lewis lung carcinoma (LL2) that spontaneously occurred, was isolated, and was derived into a cell line [32]. The LL2 cells were used to create clonally derived lines with NRF2-active (*KEAP1* KO) and NRF2-inactive (*NRF2* KO) mutations. *mKeap1* or *mNfe2l2* were knocked out using CRISPR, and 3 *KEAP1* KO and 1 *NRF2* KO clones were isolated and then validated by genomic sequencing. 

Accordingly, protein expression of NAD(P)H quinone oxidoreductase 1 (NQO1, a canonical NRF2 target gene [33]) was increased in the *KEAP1* knockout (KO) lines by ~4–7-fold (Figure 1a,b) and decreased in the *NRF2* KO clone (Figure 1c,d) by nearly 90%, along with the expected changes in NRF2 protein expression in both knockouts. NQO1 mRNA expression was also significantly (*p* < 0.01) higher in the *KEAP1* KO clones and absent in the *NRF2* KO clone (Figure 1e). Because NRF2 is a transcription factor, KEAP1 inactivation permits the activation of the NRF2 transcriptional program involving hundreds of gene targets. RNA sequencing assessed NRF2 pathway activation in the clones. 

Because of their origins as single cells, sequencing also allowed validation of transcriptomic homology between *KEAP1* KO lines. Unsupervised hierarchical clustering of *KEAP1* KO and *NRF2* KO differential gene expression compared to the LL2 parental line resulted in the clustering of *KEAP1* KO clones with each other and the *NRF2* KO clone with the LL2 parent line (Figure 1f).

Finally, QIAGEN’s IPA pathway analysis software confirmed the degree of NRF2 transcriptional activation. The Upstream Analysis feature, a method of predicting upstream regulators responsible for the differential expression in a dataset, identified NRF2 as the most highly enriched “upstream” mediator across all three *KEAP1* KO clones (*KEAP1* KO Activation Z-Scores: 4.56–5.54; *NRF2* KO Z-Score: −3.35; Appendix A). 

We further identified specific target genes regulated by NRF2 (Figure 1g). Impressively, the clones had high levels of NRF2 pathway activity with *GSTA* genes increased by a Log_2_fold-change of nearly 15 (adj. *p*: 2.76 × 10^−7^–9.17 × 10^−8^). Canonical NRF2 pathway-related genes were significantly increased in the *KEAP1* KO clones, but the same genes were either decreased or similar to WT in the *NRF2* KO clone. NRF2-regulated genes increased in the *KEAP1* KO clones include: *AKR1C* (Log_2_FC: 6.59–6.84; adj. *p*: 1.76 × 10^−4^–1.59 × 10^−5^), *CYP1A1* (Log_2_FC: 7.83–8.79; adj. *p*: 1.52 × 10^−5^–2.19 × 10^−6^), *NQO1* (Log_2_FC: 3.18–4.01; adj. *p*: 1.92 × 10^−8^–2.22 × 10^−9^), *UGT1A6* (Log_2_FC: 4.36–5.03; adj. *p*: 6.77 × 10^−5^–1.04 × 10^−5^), and *GCLM* (Log_2_FC: 2.71–3.29; adj. *p*: 4.89 × 10^−6^–4.37 × 10^−7^).

### 2.2. KEAP1 KO Enhances a Cancer-Cell Intrinsic Immunosuppressive Phenotype

To determine how *KEAP1* mutations impacted the intrinsic phenotype of the cancer lines and how they are implicated in immunosuppression, we used a multiplex assay to evaluate the secretome of the KO lines (Appendix A). The release of most cytokines was decreased in the *KEAP1* KO clones (Figure 2a). Select mediators identified by multiplex were validated using PCR (Figure 2b), which included cytokines with higher (CXCL1, CXCL5, CXCL10) and lower release (IL-33 and IL-18). Notable decreased cytokines in the *KEAP1*-KO clones included IL-7R alpha [34], IL-18 [35], IL-23 [36], IL-31 [37], and IL-33 [38], which are pro-inflammatory and correlate with infiltration and activation of T cells and anti-tumor responses. The two chemokines upregulated in *KEAP1* KO cells, CXCL1 and CXCL5 (Figure 2a,b), promote myeloid lineage infiltration into tumors [39,40,41,42,43], a significant finding because innate immune cells within the TIME suppress T cell function [44]. 

We next evaluated the expression of checkpoint markers on the cancer cells that could play a role in direct T-cell suppression. Expression of PD-L1 (*CD274*), CD80/CD86 (*CD80*), and CD155 (*PVR*) were all significantly (*p* < 0.05) higher in the *KEAP1* KO lines (Figure 2c). PD-L1 is a well-characterized immunosuppressive checkpoint maker that, when expressed on cancer cells, blocks T-cell activation through its cognate ligand PD-1 [45]. CD80 and CD86, while not as well characterized clinically, demonstrate similar immunosuppressive changes when expressed on cancer cells [46,47]. These surface proteins bind to CTLA-4, which is a clinically validated target on T cells [48]. Lastly, CD155 is the high-affinity co-receptor of TIGIT (an inhibitory T-cell receptor) and is thought to promote immunosuppression in chronically activated T cells within tumors [49,50]. The changes in cytokine release and surface marker expression were consistent with gene expression from our sequencing dataset (Figure 2d). 

Because of the known differences in immune cell accumulation in human *KEAP1*-mutant tumors [20,29] and the increased CXCL1 and CXCL5 expression suggesting increased myeloid lineage infiltration, sequencing of the knockout cells was used to compare differential expression of genes. As shown in Figure 3a, the *KEAP1* KO and *NRF2* KO clones (as compared to LL2 WT) produced distinct expression profiles. We used these sets of genes for pathway analysis to determine which overall functions are enriched in the *KEAP* KO cancer cells. Compellingly, the pathway analysis confirmed the increased expression of myeloid-recruiting chemokines (Figure 2a,b,e) and enrichment of pathways that recruit leukocytes (Figure 3b–e). Specifically, genes involved in leukocyte accumulation were significantly enriched (Figure 3b, Appendix A) in all three *KEAP* KO cancer lines (*p* = 7.24 × 10^−9^ to 1.3 × 10^−15^, Z-scores: 1.32 to 1.62), in addition to identification of a pathway specific for recruitment of neutrophils and other myeloid cells (Figure 3c, *p* = 5.95 × 10^−9^, Z-score = 2.12) in *KEAP* KO #3. Pathways involving immune cell trafficking were significantly enriched (Figure 3d,e, Appendix A) across all 3 *KEAP* KO clones but were not observed in the *NRF2* KO clone. More than 200 genes were detected in *KEAP1* KO #1, and > 300 were found in *KEAP1* KO # 2 and 3 relating to immune trafficking (Figure 3d). As compared to the *NRF2* KO line with only 10 genes, *KEAP1* KO or NRF2 pathway activation should greatly alter immune presence in tumors. Cellular recruitment and accumulation are among these trafficking processes (Figure 3e), suggesting that *KEAP1* mutations drive the recruitment of immune cells that alter the microenvironment. The most significantly enriched pathway of immune cell trafficking across the 3 *KEAP1* KO cell lines was leukocyte migration (*p* = 1 × 10^−20^ to 2.25 × 10^−34^, Appendix A), although the activation Z-score prediction varied (Z-score = −0.087 to −1.12). In addition, activation of leukocytes (*p* = 5.73 × 10^−12^ to 4.85 × 10^−21^, Z-Score = 0.38 to 0.88, Appendix A) nearly had the highest number of enriched molecules across all *KEAP1* KO lines, behind “leukocyte migration” and “cell movement of leukocytes.” For these two pathways, analysis was unable to give consistent and reliable activation Z-scores greater than 1, although their enrichment and high number of genes present support the importance of KEAP1/NRF2 in these processes. Taken together, *KEAP1* KO significantly promotes an immunosuppressive cancer cell phenotype through alterations of the secretome, regulation of surface markers, and gene expression changes consistent with modified regulation of immune trafficking.

### 2.3. KEAP1 KO Suppresses T-Cell Function in an Orthotopic Allograft Model of Lung Cancer 

Based on our hypothesis that altered immune cell recruitment can drive immune suppression, we tested whether T-cell numbers or function were decreased in a model of *KEAP1* KO lung cancer. In an orthotopic lung cancer allograft model, cancer cells were directly transplanted into the lungs of mice (Figure 4). WT, *KEAP1* KO, or *NRF2* KO lung cancer lines all formed tumors in the lungs (Figure 4a–d). We then assessed T cells that are critical for anti-tumor responses [51]. Although the overall abundance of T cells was not significantly altered (Appendix A), T-cell function was decreased. A lower percentage of cells expressing CD69 and CD107a activation markers were detected in CD8^+^ populations of *KEAP1* KO tumors (Figure 4e). CD25^+^ CD4^+^ T cells were higher in *KEAP1* KO tumors (Figure 4f), which could indicate higher Treg function [52,53]. The ratio of CD8^+^/CD4^+^CD25^+^ cells was decreased in the *KEAP1* KO tumors, although the overall CD8^+^/CD4^+^ ratio was not significantly altered (Figure 4g). This finding is relevant because a decreased CD8^+^/CD4^+^CD25^+^ ratio correlates with worse clinical outcomes in some cancers [54]. These effects on T-cell immunosuppression were not observed in *NRF2* KO tumors. Because of the variability of infiltration in immune cells using this model, we could not accurately estimate the abundance of infiltrating myeloid cells. Despite this, CD206^+^ macrophage subpopulations in *KEAP1* KO tumors were significantly increased (Figure 4h). CD206 is a marker for polarization of M2-like macrophages [55] and increased abundance of these cells within the tumor microenvironment signifies poor lung cancer prognosis [56] and immunosuppression [55,57]. Our data suggesting *KEAP1* KO cancer cells increase the recruitment of immune cells, in combination with evidence of increased CD206^+^ presence in the microenvironment, warranted further investigation.

### 2.4. KEAP1-Mutant Lung Cancer Increases Recruitment of Macrophages

To explore the cellular recruitment phenotype of cancers with *KEAP1* mutations, we turned to a publicly available single-cell sequencing dataset produced by Zavitsanou et al. [31]. In their study, *KEAP1*-mutant cell lines (*Kras^G12D^*^/+^; *p53*^−/−^ (*KP*); *KEAP1^−/−^*) were derived from a virally induced murine tumor model. These cells were injected into a tail vein, and immune cells from the resulting lung tumors sorted and sequenced. We accessed this dataset to examine the immune landscape of *KEAP1*-mutant lung tumors (Figure 5a). 

Macrophages were present in the *KEAP1*-mutant tumor model (Keap1470C) at a frequency nearly double that of Keap1WT tumors (25.9% vs. 13.5%; Figure 5b). Using this data, we subset the macrophage population and re-clustered them based on protein surface markers (antibody-derived tags) available within the sequencing dataset. The analysis identified seven clusters within the main macrophage population of pooled Keap1WT and Keap1470C tumors (Figure 5c). Importantly, clusters 2 and 3 were overrepresented in the Keap1470C macrophage population (Figure 5d). The frequency of cluster 3 was 18.95% and cluster 2 was 24.5% of overall macrophages in Keap1470C tumors, an increase of 3.5× and 3.3× respectively, compared to 5.35% and 7.4% identified in the Keap1WT tumors (Figure 5e). When the top protein markers expressed in the macrophage clusters (Figure 5f) were analyzed, each cluster had a distinct set of surface markers with cluster 2 characterized by considerably higher levels of CD38, CD14, and CD80, and cluster 3 with low MHC II (IA-IE), high F4-80, and increased CD38. Interestingly, CD206 expression was increased in both clusters 2 and 3, consistent with our findings in the lung tumor allograft model (Figure 4). 

### 2.5. Macrophage Clusters 2 and 3 Express Markers Consistent with Infiltrating Macrophages and Poor Prognosis

Next, we split the macrophage clusters by Keap1 mutational status of the tumor and assessed infiltration and prognostic markers (Figure 6). CD11b (a marker of infiltrating macrophages) was more abundant in cluster 2 and 3 macrophages (Figure 5f and Figure 6a), and accordingly, these clusters were more frequent in Keap1470C tumors with marginally higher expression of CD11b (cluster 2: Log_2_FC = 0.325, adj. *p* = 4.43 × 10^−5^; cluster 3: Log_2_FC = 0.39, adj. *p* = 1.5 × 10^−4^; Figure 6b). CD11c (a resident macrophage marker) was more highly expressed in clusters 0 and 5 (Figure 6c,d). 

Poor prognosis correlates with low macrophage MHC II [58], high CD38 expression in the tumor microenvironment [59], and high CD206 [56]. Keap1470C tumors had increased abundance of macrophage populations with detrimental expression patterns for MHC II (Figure 6e), CD206 (Figure 6g), and CD38 (Figure 6k) mostly focused within clusters 2 and 3. The number of cells with this pattern of markers was increased, but overall MHC II expression was decreased (cluster 2: Log_2_FC = −0.95, adj. *p* = 3.39 × 10^−10^; cluster 3: Log_2_FC = −0.81, adj. *p* = 4.27 × 10^−7^; Figure 6f) and CD38 slightly elevated (cluster 2: Log_2_FC = 0.45, adj. *p* = 2.88 × 10^−5^; cluster 3: Log_2_FC = 0.29, adj. *p* = 0.04; Figure 6l) in Keap1470C macrophages. 

The overall expression of CD206^+^ macrophages was not consistently altered in clusters 2 and 3 by genotype (Figure 6h). Intriguingly, in Keap1470C macrophages, CD80 was expressed at a higher level in cluster 2 and 3 (cluster 2: Log_2_FC = 0.96, adj. *p* = 5.655 × 10^−13^; cluster 3: Log_2_FC = 0.58, adj. *p* = 7.45 × 10^−5^; Figure 6i,j), which indicates an M1-like phenotype. To elaborate on this finding, we subset clusters 2 and 3 into their own respective populations and re-clustered again based on protein surface marker expression (Figure 6m,n). In cluster 2 (Figure 6m), CD80 expression was mostly present in subcluster 0, a distinct set of cells from those in subcluster 1. Strikingly, CD206 expression was nearly excluded from the CD80-high cells of subcluster 0, as almost all CD206-high cells were in subcluster 1. The same trends were observed for the populations in cluster 3 (Figure 6n). Because we had identified these high CD206-expressing cells (subclusters 1 for each macrophage cluster 2 and 3), we combined and calculated the overall number of CD206-high cells by Keap1 status. The overall percentage of CD206-high macrophages in Keap1470C tumors was 19.6%, 2.2× higher than the 8.65% in the KeapWT (Figure 6o). 

### 2.6. Prostaglandin Signaling May Mediate M2-like Macrophage Polarization in Keap1-Mutant Tumors

Finally, we sought to identify a potential mechanism by which Keap1-mutant cancers mediate inflammation and polarization of macrophages. We began by assessing the bulk-RNAseq pathway analysis using the *KEAP1* KO clones (Figure 1g and Figure 3b–d). Reassessing the data using the Diseases and Functions analysis (“downstream” functional assessment of the cancer cells) identified several pathways consistently upregulated across all the clones (Appendix A). NRF2 is a metabolic regulator with the capability of rewiring the metabolic pathways within cells [60,61,62,63,64]. Fittingly, our functional analysis identified two highly enriched pathways related to alterations in molecular fat in the three *KEAP1* KO lines: Synthesis of Lipid (Figure 7a, Z-score = 1.88–2.49) and Fatty Acid Metabolism (Figure 7b, Z-score = 0.70–3.09). 

When we assessed the top 25 enriched canonical pathways (Table 1), MIF-Regulation of Innate Immunity was significantly upregulated in all *KEAP1* KO clones (Z-score = 2.12–2.65). Genes within this canonical pathway overlap with genes found in the downstream functional analysis for Synthesis of Lipid and Fatty Acid Metabolism, such as *PLA2G2D*, *PLA2G25*, and *PTGS2* (COX-2). An excerpt from the MIF-Regulation of Innate Immunity Pathway is shown in Figure 7c. QIAGEN IPA predicts strong upregulation of this section of the pathway, implying increased prostaglandin production. Notably, high levels of prostaglandin are associated with T-cell suppression [65,66,67] and polarization to an M2-like macrophage phenotype [68,69]. We verified increased COX-2 protein level in the *KEAP1* KO cell lines (Figure 7d), which was elevated by 2–3-fold (Figure 7e), but not significantly altered in the *NRF2* KO.

The previous data regarding macrophage populations in the single-cell dataset (Figure 5 and Figure 6) exclusively evaluated protein-level surface markers. Based on the protein-marker clustering (Figure 5c), we extracted RNA gene expression data from clusters 2 and 3 and used the gene expression profiles to complete differential gene expression analysis between KeapWT and Keap1470C within each respective cluster. From this, we completed pathway analysis using the QIAGEN IPA Upstream Regulators function. In both clusters 2 and 3, prostaglandin E2 (PGE_2_) is predicted as a top upstream regulator for Keap1470C in an IPA comparison analysis between pathways for the clusters (Figure 7f). Specific genes regulated by PGE_2_ signaling found within the cluster gene expression sets are listed for cluster 2 (Figure 7g) and cluster 3 (Figure 7h). Common between both clusters, PGE_2_ activity is predicted to result in decreased *MHC* and increased *Arg1* expression, another M2-like macrophage marker.

## 3. Discussion

This study sought to identify changes in the immune microenvironment caused by cancer-specific *KEAP1* mutations. We found a cancer cell-intrinsic immunosuppressive phenotype directly caused by the loss of KEAP1 that was associated with increased recruitment and polarization of M2-like macrophages. Through creation of the clonally derived LL2 lines, we identified large changes in the cytokine secretome of *KEAP1* KO cancer cells. Cancer-derived cytokines are critical for shaping the immune microenvironment [70,71]. Therefore, it is conceivable that the broad decrease in cytokine release of *KEAP1* KO clones is partially responsible for the observed immune-cold microenvironment in human *KEAP1*-mutant cancers [29,72]. Notably, CXCL1 has a well characterized role in pancreatic ductal adenocarcinoma; it promotes myeloid cell recruitment and is associated with immunosuppressive macrophage infiltration [73]. Because CXCL1 was highly and consistently upregulated in our *KEAP1* KO cell lines, this cytokine is likely to drive the presence of M2-like macrophages in these cancers. While there was minor variability between each clone, the overall trends were similar for all *KEAP1* KO lines, with the *NRF2* KO most similar to the WT. This pattern was apparent not only in our findings of immunosuppressive phenotypes but also for the clones overall because of their hierarchical clustering patterns observed in the sequencing dataset. Generally, we observed great homology between the *KEAP1* KO cells and hypothesize some of the differences observed are the result of biological variability between cells of the LL2 WT cancer line used for their generation.

Deleterious alterations in the cytokine secretome, in combination with our sequencing data suggesting *KEAP1* mutant cells have an increased capability to regulate immune cell trafficking, led us to hypothesize that these *KEAP1*-deficient cancers are re-configuring their immune microenvironments to be more advantageous to the cancer. Decreased infiltration of lymphoid cell lineages, with some indication macrophages are also decreased, have been detected in human *KEAP1* mutant cancers [72]. However, it is still feasible that these cancers recruit new macrophage subsets and polarize them toward an immunosuppressive phenotype while following the pattern of decreased overall infiltration. To determine whether this phenomenon is relevant in human cancers, we searched the TIMER2.0 database [74,75,76] using the “mutation” immune association function, which accesses TCGA data to estimate immune cell infiltration. We identified a 3.54 Log_2_-fold increase in M2-like macrophages in *KEAP1*-mutant cancers (Appendix A). These cells were identified in squamous lung carcinomas with *KEAP1* mutations using the tumor immune dysfunction and exclusion (TIDE) analysis within the TIMER2.0 database. Unfortunately, lung adenocarcinomas are not available currently for assessment using TIDE. TIDE analysis is a computational method created to predict immune signatures in TCGA datasets. It was designed specifically to identify microenvironments associated with T-cell dysfunction and exclusion, pointedly involving the identification of M2-like macrophages associated with this phenotype [77]. These data support our hypothesis in that even though macrophages are possibly decreased overall in human *KEAP1* mutant tumors, the ones present likely have more immunosuppressive capability through their M2-polarization. 

Through our evaluation of the single-cell dataset published by Zavitsanou et al. [31], we found a higher abundance of macrophages present in Keap1R470C tumors. Intriguingly, the overall abundance of macrophages was increased, but a more striking finding was the differing compartments of identified macrophages. Macrophage clusters 2 and 3 were increased in the Keap1R470C tumors, whereas there was a relative decrease in the proportion of cluster-1 cells. Of the seven clusters identified, it is likely that clusters 0, 1, 4, and 5 represent alveolar macrophages or other resident subtypes because of the relative increase in expression of MHC II (IA-IE) and CD11c, but low CD11b expression. Clusters 2 and 3 are consistent with infiltrating macrophages, which have lower CD11c and high CD11b expression. Macrophage clusters 2 and 3 have the phenotype of infiltrating macrophages, but express relatively higher levels of CD206, the M2-macrophage marker, suggesting a pro-tumor phenotype. Because cells in these compartments also expressed CD80, canonically found on M1-like macrophages, we further evaluated subclusters within these populations and found that the cells expressing higher CD206 had much lower CD80 expression. We hypothesize that *KEAP1*-mutant tumors are recruiting macrophages to the tumor with an initial M1-like phenotype, but the cytokine and chemical milieu of the microenvironment subsequently polarizes these cells to an M2-phenotype. 

However, these findings raise questions regarding differences in macrophage infiltration between the mouse models used here and human cancers [72]. We posit that because of the broad immunosuppressive phenotype present in *KEAP1*-mutant cancers, human cancers that develop over a span of years or even decades have equilibrated to the immunosuppressive environment, which is reflected by an overall decrease in macrophages. Despite this, they retain their newly established M2-like macrophages (Appendix A) that enhance cancer immune evasion and suppress T cells. In the mouse models, which develop in the order of weeks, we may only capture the initial kinetics of *KEAP1*-mutant immune microenvironment formation led by the initial wave of M2-like macrophage polarization. This may be a potential limitation of the short-term cancer models used; nonetheless, the relevant outcomes remain similar with T-cell suppression a prevailing finding.

Consistent in many studies [20,23,41,67], *KEAP1* mutations in lung cancer result in T-cell suppression that promotes immune escape and cancer progression, a finding we also observed. Our data expand on the causes of T-cell suppression as we have further identified an increase in immunosuppressive markers expressed on *KEAP1*-mutant cancer cells that inhibit the T-cell response (PD-L1 [45], CD80/CD86 [46,47], and CD155 [49,50]). Increased immunosuppressive surface markers, along with decreased expression of T-cell promoting cytokines (IL-7R alpha [34], IL-18 [35], IL-23 [36], IL-31 [37], and IL-33 [38]) may be a source of T-cell suppression. Unfortunately, our model was not appropriate for assessing the implications of T-cell suppression on tumor growth. The LL2 model has rapid kinetics and a total timeline of less than 21 days, which is not suitable for a meaningful evaluation of tumor growth. 

In addition to differential regulation of cytokines, RNA sequencing of the cancer lines suggests a robust increase in prostaglandin synthesis pathways that could further suppress T-cell responses [65,66,67]. Prostaglandins are known immunomodulators released by various cancer types [65]. The bulk sequencing data suggesting increased prostaglandin synthesis in the *KEAP1* KO cell lines is notably strengthened by the single-cell dataset generated by Zavitsanou et al. [31], which we used to independently identify PGE_2_ signaling as a top upstream regulator of gene expression phenotypes observed in Keap1R470C macrophage clusters 2 and 3. These data suggest clusters 2 and 3 could be driven toward an M2-phenotype by PGE_2_. PGE_2_ release by cancer cells could tie together prostaglandin-mediated T-cell suppression and sustained immunosuppression by prostaglandin-driven M2-like macrophage repolarization [68,69]. In fact, the inhibition of prostaglandin signaling is being evaluated for enhancing T-cell function in cancer [78]. Although outside the scope of this current manuscript, the prostaglandin findings are an exciting avenue for future research. 

Despite the correlative nature of this data suggesting prostaglandin-mediated regulation, COX-2 overexpression alone is a negative prognostic indicator for NSCLC [79] and in pancreatic cancer correlates with T-cell exclusion [80]. COX-2/PGE_2_ signaling has many known immunomodulatory roles, and COX-2 expression generally correlates with increased PGE_2_ release [81]. Further investigation of a NRF2/COX-2/PGE_2_ axis may yield justification for combination immunotherapy treatments using COX-2 inhibitors to “sensitize” immune cells and checkpoint blockade to induce anti-cancer responses in patients bearing *KEAP1* mutations.

## 4. Materials and Methods

### 4.1. Tissue Culture

Murine Lewis Lung Carcinoma (LL2; CRL-1642-LUC2) cells were purchased from ATCC (Manassas, VA, USA) and cultured in complete DMEM media (Corning, Corning, NY, USA) containing 10% fetal bovine serum (FBS) (VWR, Radnor, PA, USA) and 1% penicillin and streptomycin (Corning). All cells were used within eight passages after thawing and cultured under a humidified atmosphere at 37 °C with 5% CO_2_. 

### 4.2. Cell Line Creation

LL2 knockout cell lines were created using a Lonza Nucleofection kit using plasmids containing CRIPSR components and guide RNAs. 4 × 10^6^ cells were spun at 90 *g* for 10 min, resuspended in Lonza SE Nucleofection Buffer (Lonza, Basel, Switzerland) and split into two tubes containing 2.5 μg of plasmid vector. Plasmids carried a gRNA for mKeap1 [VB900065-3915qbt] or mNfe2l2 [VB900051-0812sph], GFP, and the Cas9 enzyme (VectorBuilder, Chicago, IL, USA). The cell suspensions were loaded into Nucleofection cuvettes and protocol DS-150 run to facilitate plasmid entry (optimized protocol for this cell line) using a Lonza-4D Nucleofector. Cells were allowed to rest for 10 min at room temperature prior to adding 400 μL of complete cell media. Recovered cells were plated in 5 cm dishes with 5 mL of media and allowed to rest for 24 h. Single, viable (DAPI-), GFP+ clones were sorted on a BD FACS Aria (BD Biosciences, Franklin Lakes, NJ, USA) into a 96-well plate containing 100 μL of conditioned cell media (42% LL2 cell-conditioned media, 42% fresh complete DMEM, 16% FBS). Single clones were expanded for 3–4 weeks until sufficient cells were available for freezing and validation by Western blotting. Clones with evidence of KO by Western blotting were sent to Azenta Life Sciences (Burlington, MA, USA) for independent confirmation of genomic editing by Sanger sequencing.

### 4.3. Western Blotting 

LL2 cells were plated in 10 cm dishes and cultured for 48 h in complete media. Protein was harvested into RIPA buffer (5 M NaCl, 1 M Tris-Cl, 0.5 M EDTA, 25 mM deoxycholic acid, 1% Triton X-100 (*v*/*v*), 0.1% sodium dodecyl sulfate (*w*/*v*), pH 7.4) containing protease inhibitors (1 mM phenylmethylsulfonylfluoride, 2 mg/mL aprotinin, and 5 mg/mL leupeptin) purchased from Millipore-Sigma (Burlington, MA, USA). A bicinchoninic acid assay (Millipore-Sigma) was used to determine protein concentration, and 30 µg of protein was separated by gel electrophoresis on 10% SDS-PAGE gels. Proteins were then transferred to nitrocellulose membranes and blocked with 5% bovine serum albumin (*w*/*v*) in tris-buffered saline containing 0.1% tween-20 (*v*/*v*) for 1 h. Primary antibodies were incubated overnight at 4 °C [anti-NRF2 (Novus Biologicals, Centennial, CO, USA, 1:1000), anti-KEAP1, anti-NQO1, anti-COX-2 (Cell Signaling Technology, Danvers, MA, USA, 1:1000), and anti-actin (Cell Signaling Technology, 1:7000)] and secondary antibodies [anti-rabbit CW800 (Li-Cor, Lincoln, NE, USA, 1:10,000) or anti-mouse CW800 (Li-Cor, 1:10,000)] at room temperature for 1 h. Bands were detected using a Li-Cor CLx machine and band intensity quantified using Image Studio Lite.

### 4.4. Bulk RNA Sequencing

Confluent LL2 cell lines were split 1:4 into 10 cm dishes and cultured for 24 h in complete media. RNA was isolated using a RNeasy MiniPrep Kit (QIAGEN, Hilden, Germany). Total RNAs with a 260/280 ratio > 2.0 and 260/230 ratio > 1.9 were submitted in biologic triplicates to the Van Andel Research Institute Genomics Core, which prepared libraries from 500 ng of total RNA using the KAPA mRNA Hyperprep kit (v4.17, Kapa Biosystems, Wilmington, MA, USA). RNA was sheared to 300–400 bp. Prior to PCR amplification, cDNA fragments were ligated to IDT for Illumina TruSeq UD Indexed adapters (Illumina Inc., San Diego, CA, USA). The quality and quantity of the finished libraries were assessed using an Agilent DNA High Sensitivity chip (Agilent Technologies, Inc., Santa Clara, CA, USA), QuantiFluor^®^ dsDNA System (Promega Corp., Madison, WI, USA), and Kapa Illumina Library Quantification qPCR assays (Kapa Biosystems). Individually indexed libraries were pooled and 50 bp paired end sequencing was performed on an Illumina NovaSeq6000 sequencer using an S2, 100 bp sequencing kit (Illumina Inc.) to an average depth of 30 M reads per sample. Base calling was done by Illumina RTA3 and output of NCS was demultiplexed and converted to FASTQ format with Illumina Bcl2fastq v1.9.0. Raw read counts were received as FASTQ files, and data processed according to the workflow by Berry et al. [82]. In brief, pseudo-alignment to the Mus Musculus GRCm38.p6 cDNA genome was completed with Kallisto v0.48.0 [83] read-mapping software and low read counts (<3 counts per million per gene summed across all samples) were subsequently filtered from the dataset. Identification of differential gene expression was determined using standard processing with the Limma v3.58.1 Bioconductor package for R [84,85,86]. Networks, canonical pathways, and functional analyses were analyzed using QIAGEN IPA software. Raw data was deposited on the GEO and can be accessed through accession GSE250166.

### 4.5. Protein Multiplex

LL2 clones (1 × 10^6^ cells) were plated in 5 cm dishes in serum-deprived (1% FBS) media. After 48 h, conditioned media was collected and cleared by centrifugation. Samples were snap-frozen in liquid nitrogen and kept at −80 °C until use. Secreted proteins were evaluated using the ProcartaPlex Mouse Immune Monitoring Panel 48-plex (ThermoFisher Scientific, Waltham, MA, USA). On the day of analysis, samples were thawed on ice, prepared, and data acquired on a Luminex 200 (Luminex Corporation, Austin, TX, USA) instrument according to the manufacturer’s protocol.

### 4.6. RT-qPCR

A total of 6 × 10^5^ LL2 cells were plated in complete media and allowed to adhere for 24 h. RNA was isolated using TRIzol regent (ThermoFisher Scientific) and 1 µg of RNA was converted to cDNA using a High-Capacity cDNA Reverse Transcription Kit (Applied Biosystems, Waltham, MA, USA) according to the manufacturer’s protocol. Primers for each target were purchased from IDT (Coralville, IA, USA). Fast SYBR Green Master Mix (Applied Biosystems) was used to run samples on the QuantStudio7 Flex RT-PCR system (Applied Biosystems) and relative gene expression calculated by normalizing CT values to GAPDH and vehicle (DMSO) control using ΔΔCT. 

### 4.7. Cancer Line Surface Marker Staining

LL2 cells (5 × 10^5^) were harvested, washed, and stained in tubes. Zombie Aqua was used as a viability stain (BioLegend, San Diego, CA, USA). The antibody cocktail included anti-PD-L1/PE; anti-CD80/BV650; anti-CD86/BV785; anti-CD155/PE-Cy7; and Viability/Zombie Aqua, all from BioLegend. Cells were stained in Brilliant Violet staining buffer (BioLegend) on ice for 30 min, fixed (Fixation buffer, BioLegend) and washed. Data was acquired on an Attune Cytpix (ThermoFisher). Final data were analyzed using FlowJo v10.9.0 (BD Biosciences). Fluorescence minus one (FMO) samples were used for identifying positive signals. Median fluorescence intensity for each cell marker was evaluated by FlowJo and computed as a percentage increase from LL2 WT cell line for each knockout cell line. 

### 4.8. Mice and Orthotopic Injection Model

C57BL/6 mice purchased from Jackson Labs (Bar Harbor, ME, USA) were bred in-house at Michigan State University (MSU). All experiments were approved by the Institutional Animal Care and Use Committee at MSU. LL2 WT, *KEAP1* KO, or *NRF2* KO cell lines (6000 cells/injection) were orthotopically injected in the lungs of 3 cohorts of 8-week-old male C57BL/6 mice. Mice were anesthetized with isoflurane and a 30 g needle was advanced through the skin of the 4th or 5th intercostal space in the left parasagittal line on the dorsal side and into the lung for injection of the cell suspensions in a 1:1 saline:Matrigel mixture. Tumors were allowed to grow for 21 days and then either imaged using IVIS or harvested and dissociated into single-cell suspensions, stained, and evaluated by flow cytometry. For IVIS imaging, hair was removed using Nair hair removal cream 24 h before image capture. Mice given intraperitoneal injections of D-luciferin (150 mg/kg of body weight) were anesthetized with isoflurane, then the total luminescence was captured in vivo after 10 min and in lungs ex vivo immediately thereafter. Lung tumors were harvested and stained with Hematoxylin and Eosin by the MSU Investigative Histopathology Core. 

### 4.9. Immunophenotyping

The left tumor-bearing lung was isolated and digested in FBS-free DMEM media containing collagenase (300 U/mL, Millipore-Sigma) and DNAse (2 U/mL, Calbiochem, San Diego, CA, USA) for 30 min at 37 °C with stirring. Samples were strained through a 40 µm cell filter and red blood cells lysed using RBC lysis solution (150 mM NH_4_Cl, 14.1 mM NaHCO_3_, 0.1 mM EDTA, pH 7.3; chemicals from Millipore-Sigma). Prior to staining, cells were treated with anti-CD16/CD32 antibody (BioLegend) to block unoccupied Fc receptors and lower background staining. Single-cell solutions were resuspended in Brilliant Violet Buffer (BD Biosciences) and stained for 30 min on ice using an optimized antibody panel [87]. Stained cells were fixed in Fixation Buffer (BioLegend) for 20 min on ice and washed prior to running on a cytometer. Data were acquired on a five-laser Cytek Aurora System using SpectroFlo software v2.2 (Cytek Biosciences, Fremont, CA, USA). Final data were analyzed using FlowJo v10.9.0 (BD Biosciences) using a previously published gating strategy [87]. 

### 4.10. Single-Cell Analysis 

Single-cell analysis was completed using published data [31] and their publicly accessible dataset through the Gene Expression Omnibus [88,89] (GSE2414820). Data were analyzed as described by Berry et al. using open-source software [82] and the R Studio v4.2.1 development environment [90]. The Seurat v5.0.1 R package was used for data integration, UMAP dimensionality reduction, and differential gene expression of datasets [91]. A Seurat object containing metadata regarding the original data processing, normalization, and cellular identification of broad cell clusters (i.e., T cells, NK cells, macrophages, etc.) was downloaded directly from GEO servers. Because this data had already been processed into a useable form by Zavitsanou et al. [31], it was not further modified or normalized. The downloaded file also contained multimodal Seurat assay data for RNA gene expression and “ADT” antibody-derived tags with protein-level information (CITE-Seq) on cell surface markers. Cluster identification and surface marker characterization were completed using CITE-Seq protein-level information. The FindNeighbors and FindClusters functions were used after RunUMAP to re-cluster and identify macrophage populations. FindAllMarkers was used to identify differential expression (ADT) between all macrophage subclusters. Differential gene expression comparing Keap1470C and Keap1WT in macrophage clusters 2 and 3 was completed using RNA gene expression data and the FindMarkers function within Seurat. The differential expression was uploaded into the QIAGEN IPA software for pathway analysis. 

### 4.11. QIAGEN Ingenuity Pathway Analysis Software

QIAGEN Ingenuity Pathway Analysis (IPA) Software, Version: 107193442 [92] was used for all gene expression pathway analysis using bulk and single-cell sequencing datasets (QIAGEN Inc., https://digitalinsights.qiagen.com/IPA). Analyses were restricted to Log2-fold changes > 1 and adjusted *p*-values < 0.05 for bulk sequencing, and Log2-fold changes > 0.75 and *p*-values < 0.05 in the single-cell dataset. Functions used within the IPA software include Canonical Pathways, Upstream Regulators, and Diseases and Functions for each independently computed differential expression dataset. Comparison analyses were used to combine *KEAP* KO clones, as well as macrophage clusters 2 and 3 using default settings. 

### 4.12. Statistical Analysis

All statistical analyses used GraphPad Prism 10 software, R Studio with the indicated R package, or were computed by QIAGEN’s IPA software. Hypothesis testing was completed using the appropriate statistical test. For three or more groups, a one-way ANOVA with Dunnett post-hoc was used. For two groups, a *t*-test was used. If the data failed a Brown–Forsythe test for equal variance, a non-parametric test was used: Kruskal-Wallace with Dunn’s multiple comparisons in place of ANOVA or Mann–Whitney in place of a *t*-test. Limma (bulk-sequencing) uses empirical Bayes statistics for hypothesis testing. Statistical outputs of the Seurat package (single cell) are based on the non-parametric Wilcoxon rank sum test. Within Seurat, the DESeq2 package uses the likelihood ratio test for hypothesis testing of differentially expressed genes between three or more groups. Alpha was set to (α = 0.05) for all applications.

## 5. Conclusions

In summary, we have identified deleterious cancer-cell intrinsic changes facilitated directly by *KEAP1* KO. These include broad decreases in cytokine release, increased cancer-promoting chemokines (CXCL1 and CXCL5) and increased immunosuppressive surface proteins that likely facilitate T-cell suppression. We further identified the capability of *KEAP1* KO cells to regulate immune cell infiltration and polarization within tumors, particularly in promoting an M2-like macrophage phenotype. Finally, we propose that prostaglandin release by *KEAP1* KO cancer cells warrants further investigation as a mechanism for driving T-cell suppression and polarization of immunosuppressive macrophages. Our future directions include validation of prostaglandin signaling in human tumors, assessing direct M2 polarization by PGE_2_ derived from *KEAP1* KO cancer cells, and assessment of the overall TIME in human NSCLC. 

## Figures and Tables

**Figure 1 ijms-25-03510-f001:**
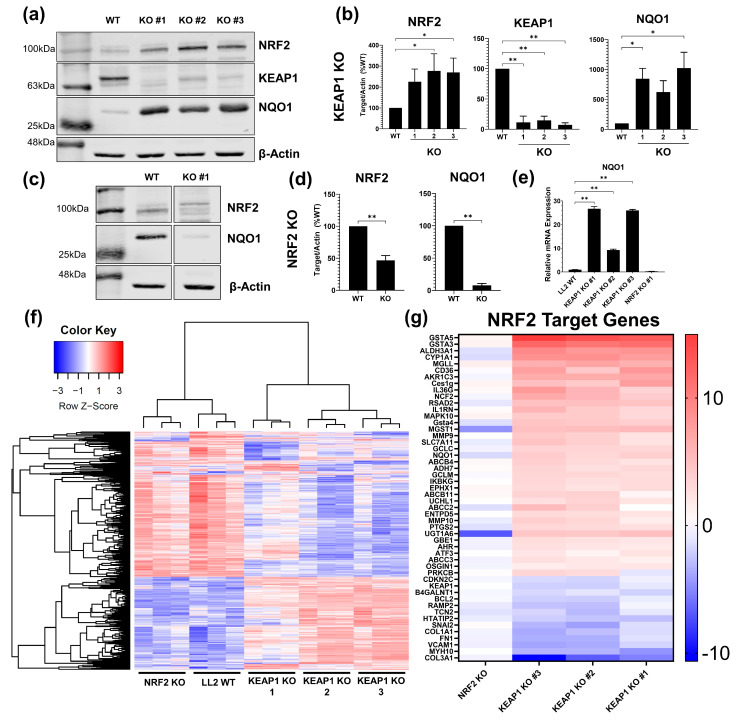
Creation and characterization of Kelch-like ECH-associated protein 1 (*KEAP1)* and Nuclear factor erythroid 2-related factor 2 (*NRF2)* knockout (KO) clones. LL2 cells were transfected with CRISPR plasmids targeting *mKeap1* or *mNfe2l2*, then single clones sorted and expanded into cell lines. Knockout of the selected proteins and expression of the NRF2 downstream target protein NQO1 were validated by Western blotting. Representative Westerns for *KEAP1* and *NRF2* knockout clones are shown in (**a**,**c**)**,** with quantification of band intensity in (**b**,**d**), respectively. β-actin was included as loading control; (**b**,**d**) are graphed as a percentage of WT band intensity ratio of target gene/β-actin. Expression of *NQO1* mRNA (**e**) was measured by RT-qPCR and calculated using the ΔΔCT method comparing target gene expression to both LL2 WT and the *GAPDH* housekeeping gene. (**f**) Hierarchical clustering from RNA sequencing of *KEAP1* and *NRF2* KO differential gene expression compared to LL2 WT. (**g**) Target genes of NRF2 differentially expressed in the *KEAP1* and *NRF2* KO sequencing datasets were identified using the upstream pathway analysis function of QIAGEN’s IPA Software. * = *p* < 0.05, ** = *p* < 0.01. n = 3–4 independent experiments for Western and RT-PCR data. n = 3 biologic replicates/cell line in RNA sequencing.

**Figure 2 ijms-25-03510-f002:**
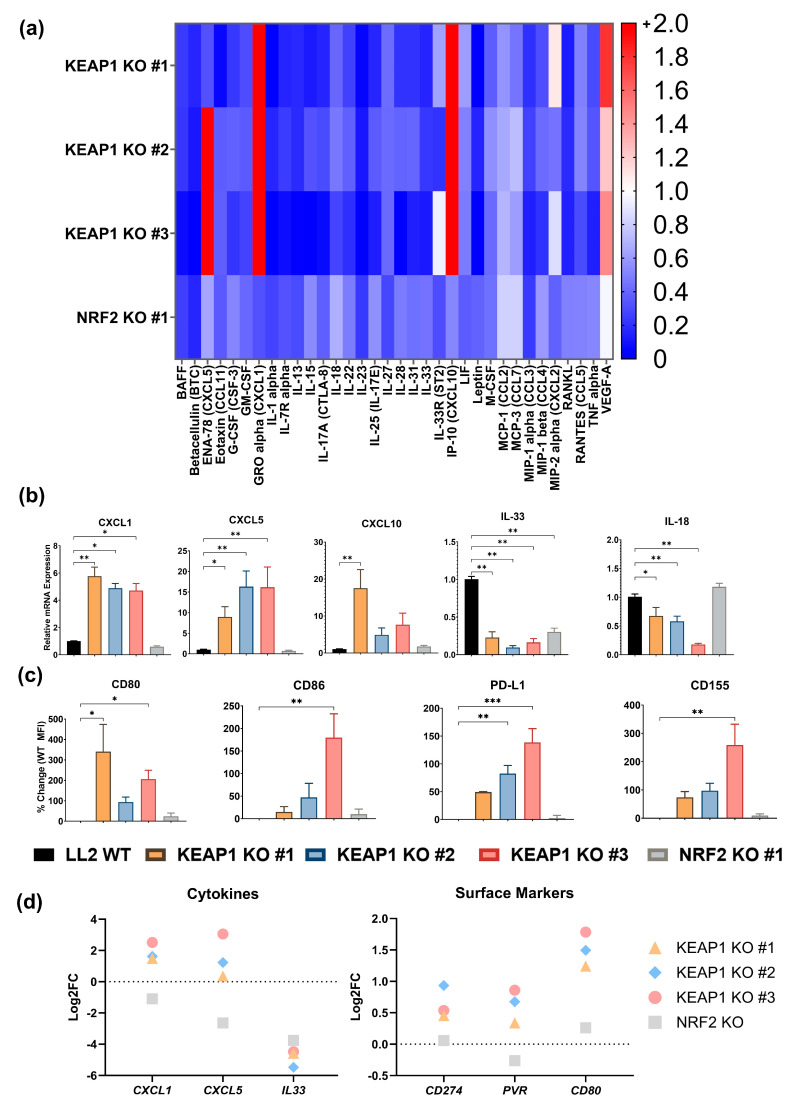
*KEAP1* KO increases the expression of myeloid-recruiting chemokines and immunosuppressive surface markers. (**a**) Secretion of cytokines in the supernatants by LL2 *KEAP1* and *NRF2* KO clones was assessed using a protein multiplex assay. Data are graphed as fold change compared to LL2 WT secretion. (**b**) PCR validation of select cytokines in (**a**). The ΔΔCT method was used to compare target gene expression to both LL2 WT and the *GAPDH* housekeeping gene. (**c**) Immunosuppressive surface marker expression in the *KEAP1* and *NRF2* KO clones was measured by flow cytometry. Data are graphed as a percentage change in MFI compared to LL2 WT. (**d**) Log_2_-Fold changes of surface markers and cytokines assessed in (**a**–**c**), calculated using the RNA sequencing dataset in Figure 1. Adjusted *p*-value < 0.05 for at least one *KEAP* KO group per gene. * = *p* < 0.05, ** = *p* < 0.01, *** = *p* < 0.001. n = 2–3 biologic replicates or independent experiments.

**Figure 3 ijms-25-03510-f003:**
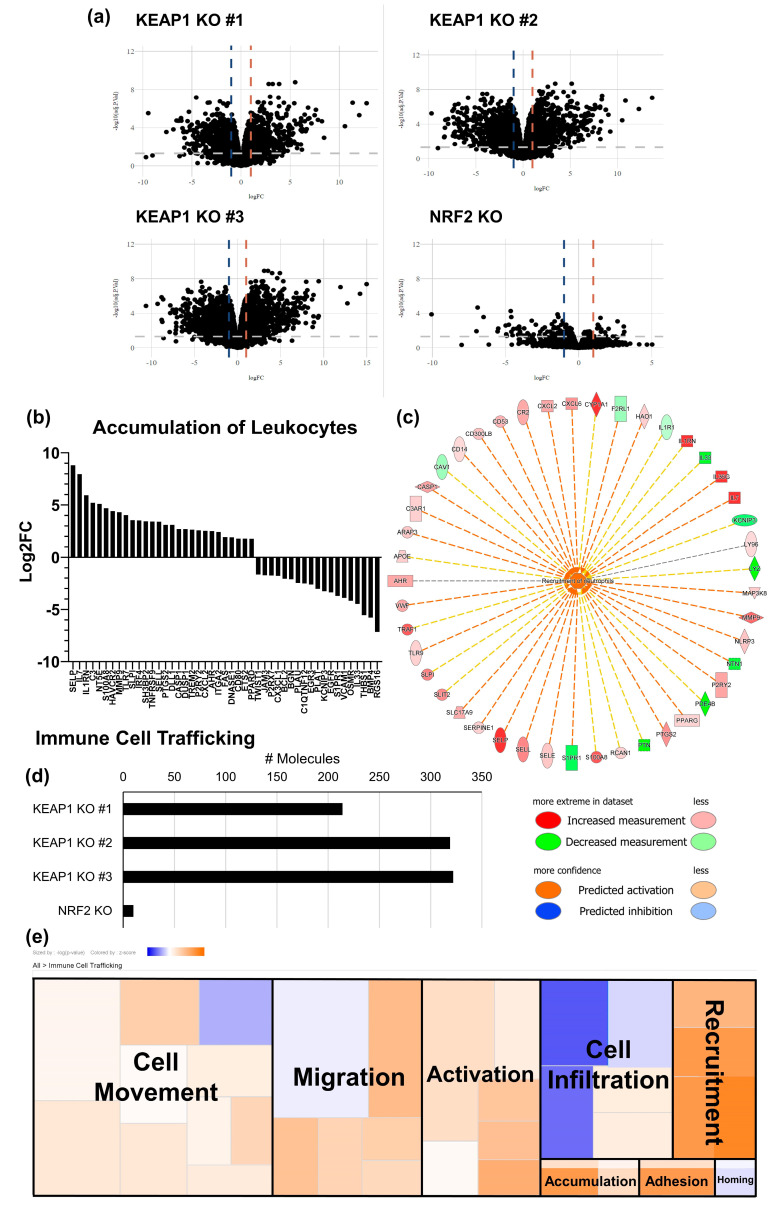
Expression of genes involved in the recruitment and accumulation of leukocytes are enriched in *KEAP1* KO clones. *KEAP1* and *NRF2* KO cell lines were analyzed by RNA sequencing to assess gene expression profiles. (**a**) Volcano plot of differentially expressed genes in each cell line compared to LL2 WT. Vertical dashed lines: Log_2_-fold change = −1 (blue) or 1 (orange), horizontal dashed line: adjusted *p*-value = 0.05. (**b**) Genes involved in the accumulation of leukocytes identified by QIAGEN’s IPA software using differential gene expression input. Data shown are from *KEAP1* KO #3, representative of all *KEAP1* KO clones. (**c**) Diagram of a gene set enriched in *KEAP1* KO #3 for the recruitment of neutrophils with a Z-score > 2. Orange dashed lines indicate gene measurement direction is consistent with increased function, yellow dashed lines signify an inconsistent measurement, and grey dashed lines indicate a known association, but unknown directionality based on existing literature within the IPA database. (**d**) The number of differentially expressed genes detected for each cell line with known functions in immune cell trafficking. (**e**) QIAGEN IPA predictions of functional changes in pathways involved in immune cell trafficking. The figure for *KEAP1* KO #3 is shown. Orange—increase, blue—decrease, color intensity—magnitude of predicted change.

**Figure 4 ijms-25-03510-f004:**
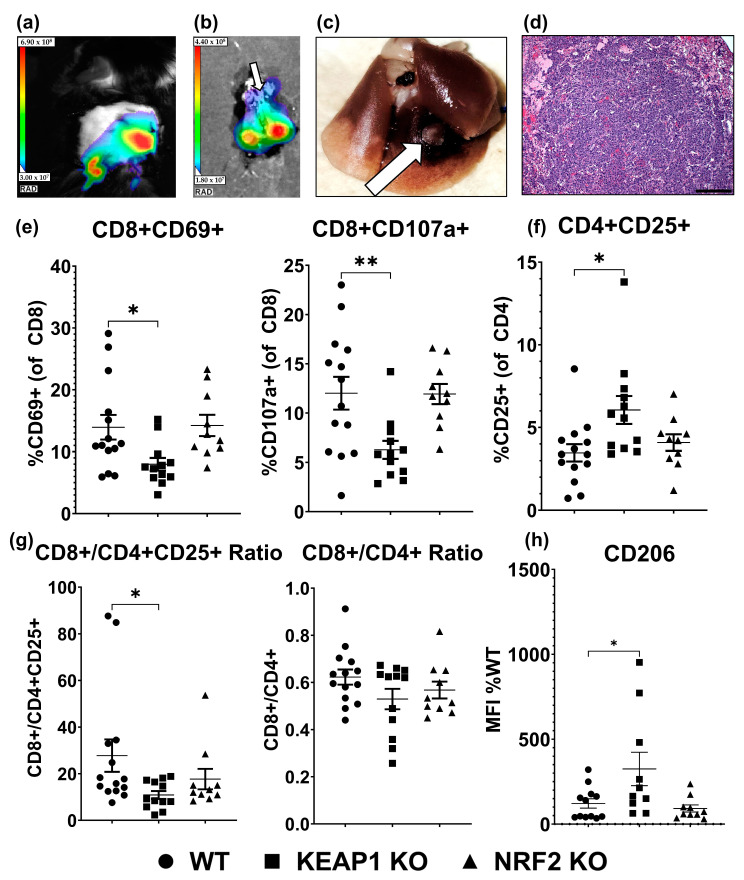
Low expression of T-cell activation markers but high expression of M2-like macrophage markers in *KEAP1*-mutant tumors. LL2 lung cancer cells were injected directly into the lungs of C57BL/6 mice and allowed to grow for 21 days. Tumor formation in this model was confirmed in several mice injected with WT cells (**a**–**d**). (**a**) In vivo and (**b**) ex vivo luciferase imaging confirmed the presence of tumors. The white arrow in (**b**) orients to the trachea. (**c**) Gross tumor morphology with white arrow pointing toward a tumor within the lung. (**d**) Hematoxylin and Eosin staining of WT tumor. Scale bar = 300 μm. T-cell activation status in LL2 WT, *KEAP1* KO, and *NRF2* KO orthotopic tumors were evaluated by flow cytometry (**e**–**g**). (**e**) Percentage of cells expressing activation markers in the CD8^+^ lineage. (**f**) Percentage of CD25^+^ cells in the CD4^+^ lineage. (**g**) The ratios of CD8^+^/CD4^+^CD25^+^ and CD8^+^/CD4^+^ cells were calculated from their respective percentage of CD45^+^ cells in the tumors. (**h**) Infiltrating macrophage CD206 MFI as a percentage of WT MFI. Each point represents one mouse. Data are mean +/− S.E. n = 10–14 mice/group. * = *p* < 0.05. ** = *p* < 0.01.

**Figure 5 ijms-25-03510-f005:**
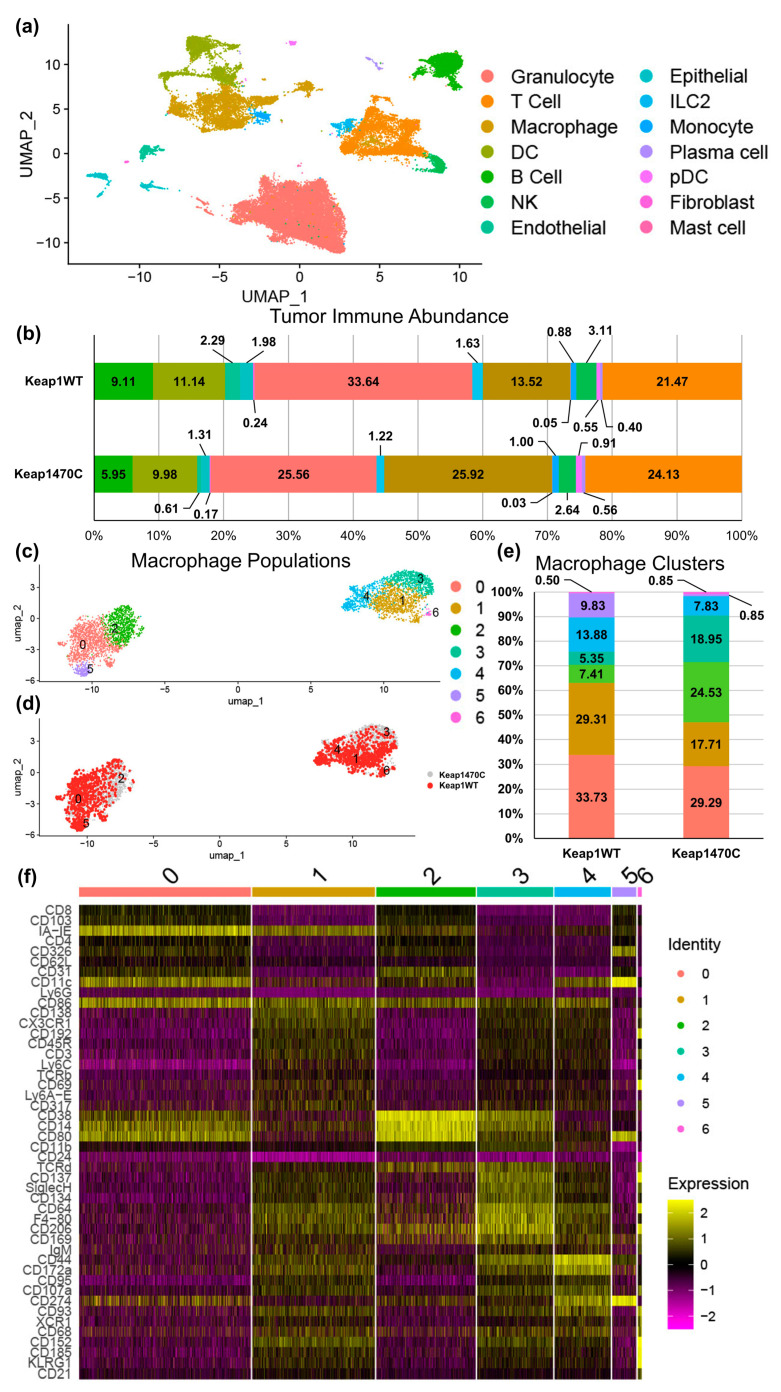
High macrophage abundance in murine lung tumors with Keap1R470C mutations. Single-cell data generated by Zavitsanou et al. [31] compared KeapWT and Keap1R470C-mutant (Keap1470C) cells from a Kras^G12D/+^; p53^−/−^ (KP) tail-vein model of murine lung cancer. (**a**) UMAP clustering of all combined lung tumors in the dataset colored by cell type. (**b**) Relative immune cell abundances for KeapWT and Keap1470C mutant tumors based on UMAP clustering cell identification. Colors in (**b**) correspond to the legend in (**a**). The percentage abundance of each cell type is overlayed on the histogram. (**c**) UMAP re-clustering of cells identified as macrophages in (**a**) colored by new cluster ID. (**d**) UMAP in (**c**) colored by tumor mutational status. (**e**) Relative abundance of clusters identified in (**c**) for KeapWT and Keap1470C tumors. (**f**) Heat map of top 10 identifying markers of each cluster identified in (**c**). UMAP dimensionality reduction was followed by weighted-nearest neighbor analysis, both using input from ADT-sequenced protein surface marker expression. (**a**,**b**) These figures were re-generated using our own analyses with permission from Elsevier, license number 5686531055863.

**Figure 6 ijms-25-03510-f006:**
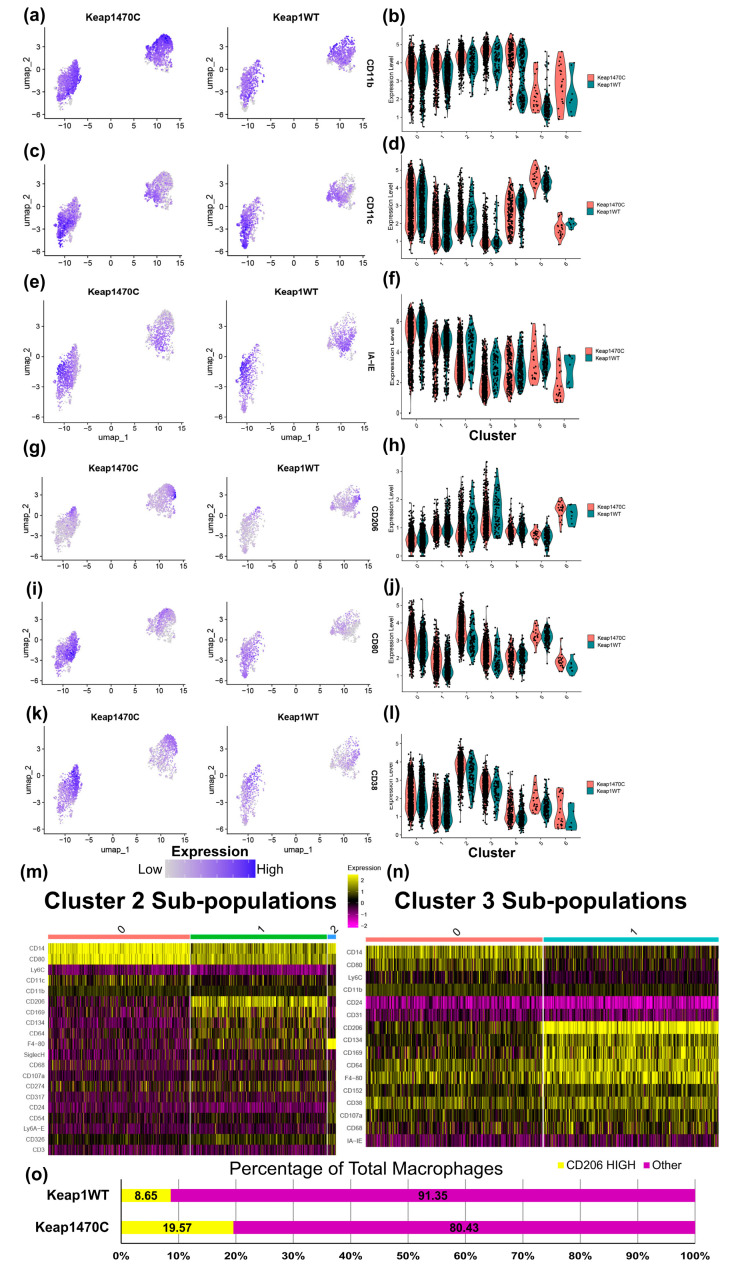
Macrophage clusters 2 and 3 express markers consistent with infiltrating M2-like macrophages. Dot plots overlayed with surface marker expression generated from the single-cell dataset, and corresponding violin plots of relative marker expression across all clusters, split by tumor genotype (**a**,**b**) CD11b, (**c**,**d**) CD11c, (**e**,**f**) IA-IE, (**g**,**h**) CD206, (**i**,**j**) CD80, and (**k**,**l**) CD38. (**m**,**n**) Sub-setting of macrophage clusters 2 and 3 followed by re-clustering into subpopulations yielded new subclusters. Heat maps of top 10 identifying markers expressed in the newly identified subclusters in (**m**) cluster 2 and (**n**) cluster 3. (**o**) Overall abundance of “CD206-High” clusters (cluster 2, subcluster 1 and cluster 3, subcluster 1) as a percentage of total macrophages identified in each tumor genotype. Assessment of surface markers present on macrophage cluster 2 and 3 subclusters from ADT-sequenced protein data. UMAP dimensionality reduction was followed by weighted-nearest neighbor analysis, both using input from ADT-sequenced protein surface marker expression.

**Figure 7 ijms-25-03510-f007:**
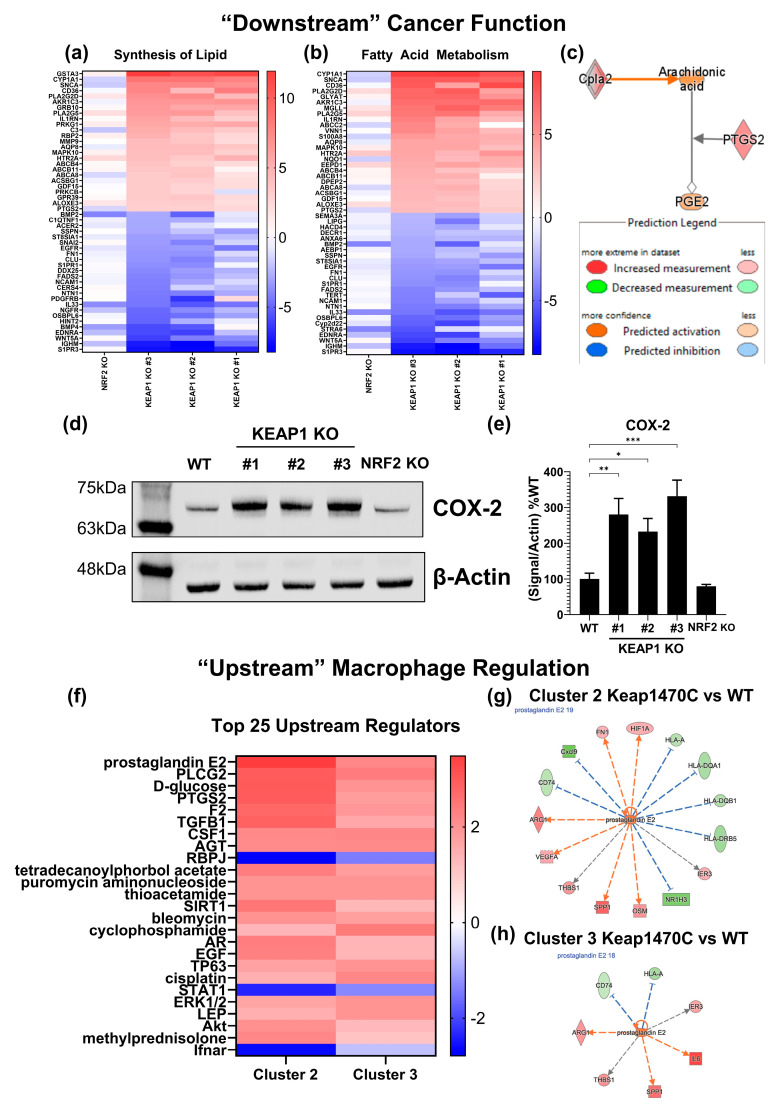
Prostaglandin signaling predicted to contribute to M2-like macrophage polarization in *Keap1*-mutant cancers. (**a**–**c**) QIAGEN IPA “Downstream” pathway analysis using LL2 cancer line sequencing. Heatmap of the top 50 genes for enriched pathways: (**a**) Synthesis of Lipid and (**b**) Fatty Acid Metabolism from *KEAP1*-KO and *NRF2* KO differential gene expression datasets (Figure 1 and Figure 3). (**c**) Excerpt from the “MIF-Regulation of Innate Immunity” canonical pathway focusing on prostaglandin synthesis. Overlayed gene expression was measured in *KEAP1* KO #3, representative of all *KEAP1* KO clones. Representative Western blot for protein validation (**d**) and relative quantification (**e**) of COX-2 (*PTGS2*) expression in LL2 cell lines; (**e**) is graphed as a percentage of WT band intensity ratio of target gene/β-actin. (**f**–**h**) QIAGEN IPA Upstream analysis using cluster 2 and 3 differential gene expression comparing Keap1R470C to KeapWT. (**f**) Top 25 upstream regulators common to both clusters identified by the IPA software in a comparison analysis. Genes known to be regulated by prostaglandin E2 differentially expressed in Keap1R470C tumor macrophages of (**g**) cluster 2 and (**h**) cluster 3. Gene expression is overlayed on each molecule. Coloring corresponds to the legend in (**c**). Differential gene expression and pathway analysis for clusters 2 and 3 used mRNA-level data. * = *p* < 0.05, ** = *p* < 0.01, *** = *p* < 0.001. n = 4 independent experiments for Western data.

**Table 1 ijms-25-03510-t001:** Top 25 Differentially Regulated Pathways in KEAP1 and NRF2 KO Cell Lines.

QIAGEN IPA Activation Z-Score Prediction				
	*NRF2 KO*	*KEAP1 KO #3*	*KEAP1 KO #2*	*KEAP1 KO #1*
Xenobiotic Metabolism AHR Signaling Pathway	−2.449	2.309	2.496	2.53
Pulmonary Fibrosis Idiopathic Signaling Pathway		−3.677	−3.576	−2.191
Glutathione-mediated Detoxification	−2.236	2.646	2.646	1.633
G Beta Gamma Signaling		−3.153	−3	−2.646
Xenobiotic Metabolism General Signaling Pathway	−2.449	2.673	1.886	1.604
Role of NFAT in Cardiac Hypertrophy		−3.13	−3.128	−2.183
CREB Signaling in Neurons		−2.562	−3.048	−2.722
Myelination Signaling Pathway		−2.469	−3.111	−2.746
Acetylcholine Receptor Signaling Pathway		−2.858	−2.414	−2.683
ID1 Signaling Pathway		−2.449	−3	−2.496
Dermatan Sulfate Biosynthesis (Late Stages)		−2.646	−2.828	−2.449
Chronic Myeloid Leukemia Signaling	2	−1.667	−3.202	−0.943
Estrogen Receptor Signaling		−2.191	−2.828	−2.683
GPCR-Mediated Nutrient Sensing in Enteroendocrine Cells		−2.84	−2.324	−2.53
MIF-mediated Glucocorticoid Regulation		2.646	2.449	2.449
Netrin Signaling		−2.53	−2.714	−2.236
Hepatic Fibrosis Signaling Pathway		−2.714	−3.286	−1.461
Cardiac Hypertrophy Signaling (Enhanced)	0	−2.48	−2.832	−2.137
Chondroitin Sulfate Biosynthesis (Late Stages)		−2.236	−2.646	−2.449
WNT/Ca+ pathway		−2.53	−2.673	−2.121
Role of Osteoclasts in Rheumatoid Arthritis Signaling Pathway		−2.795	−2.777	−1.569
MIF Regulation of Innate Immunity		2.333	2.121	2.646
Transcriptional Regulatory Network in Embryonic Stem Cells		−2.065	−2.646	−2.333
Factors Promoting Cardiogenesis in Vertebrates	0	−2.6	−3.024	−1.414
NRF2-mediated Oxidative Stress Response		3.153	1.807	1.732

## Data Availability

All data and project resources will be made available by request, including *KEAP1* KO and *NRF2* KO cell lines, in accordance with the NIH Grant Policy on Sharing of Unique Research Resources (https://grants.nih.gov/grants/policy/nihgps/HTML5/section_8/8.2.3_sharing_research_resources.htm, accessed on 16 March 2024). Bulk RNA gene expression has been deposited on the Gene Expression Omnibus, a public repository sponsored by the NIH. It can be accessed through (GSE250166) after the date of publication.

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
