# Peer review of "KEAP1-Mutant Lung Cancers Weaken Anti-Tumor Immunity and Promote an M2-like Macrophage Phenotype"

_ijms, 2024, doi:10.3390/ijms25063510_

Round 1

Reviewer 1 Report

Comments and Suggestions for Authors

The authors carried out an important study, non-trivial in terms of goals and objectives, aimed at studying the role of NRF2 activation in cancer cells for the development of optimal treatment regimens for lung cancer in the future. To achieve their goals, the authors obtained lung cancer cell lines in mice edited using CRISPR by excluding the KEAP1 or NFE2L2 genes, which in itself is a very interesting procedure. In addition, it was possible to detect harmful internal changes in cancer cells, including a significant decrease in cytokine release, an increase in cancer-promoting chemokines, and an increase in immunosuppressive surface proteins that likely contribute to T-cell suppression. I believe that the experiment deserves attention and in the future may form the basis for a more complete and expanded study. As a note, I would like to ask the authors, in conclusion, to more broadly describe the prospects for the development of this research.

Author Response

We thank the reviewer for the opportunity to expand upon future directions of this research. We have added additional discussion regarding the future prospects to the end of the discussion section and to the conclusions section. As noted in the final paragraph of the manuscript, “In summary, we have identified deleterious cancer-cell intrinsic changes facilitated directly by KEAP1 KO. These include broad decreases in cytokine release, increased cancer-promoting chemokines (CXCL1 and CXCL5) and increased immunosuppressive surface proteins that likely facilitate T cell suppression. We further identified the capability of KEAP1 KO cells to regulate immune cell infiltration and polarization within tumors, particularly in promoting an M2-like macrophage phenotype. Finally, we propose that prostaglandin release by KEAP1 KO cancer cells warrants further investigation as a mechanism for driving T cell suppression and polarization of immunosuppressive macrophages. Our future directions include validation of prostaglandin signaling in human tumors, assessing direct M2 polarization by PGE2 derived from KEAP1 KO cancer cells, and assessment of the overall TIME in human NSCLC.”

Reviewer 2 Report

Comments and Suggestions for Authors

The paper could be accepted after minor revisions. 

I am not sure but probably Conclusions should be in the paper. In this case, the paragraph from lines 472-479 could be moved to Conclusions. Types of cytokines should be specified in Conclusions. 

Some sections should be presented in several paragraphs, because the text hard to read in form of one paragraph. For example, in section 2.1 the paragraphs could starting from Lines 102, 106, 110. Same for sections 2.4 (starting new paragraph from line 250) and section 2.5 (starting new paragraphs from lines 286 and 294), and section 2.6 (starting from lines 334, lines 342). 

Please add a paragraph's space on lines 266, 311, 356. 

Figure 5, 6 (line 343) and Figure 5c (line 344) should be in bold. 

In Figure 2b there is no titles on OY axis in 4 figures in the line. So, please, remove also titles on OY axis in figures in line with same titles in Figure 1b and Figure 2c.

Author Response

  • I am not sure but probably Conclusions should be in the paper. In this case, the paragraph from lines 472-479 could be moved to Conclusions.
    • We thank the reviewer for their comment and have added a conclusions section to our paper.
  • Types of cytokines should be specified in Conclusions. 
    • Our thanks for noting this point of clarification and the cytokines have been specified in the new conclusions section.
  • Some sections should be presented in several paragraphs, because the text hard to read in form of one paragraph. For example, in section 2.1 the paragraphs could starting from Lines 102, 106, 110. Same for sections 2.4 (starting new paragraph from line 250) and section 2.5 (starting new paragraphs from lines 286 and 294), and section 2.6 (starting from lines 334, lines 342). 
    • This is an important comment to increase the ease of reading our paper. We have made the suggested changes.
  • Please add a paragraph's space on lines 266, 311, 356. 
    • We thank the reviewer for identifying this formatting error and the change has been corrected.
  • Figure 5, 6 (line 343) and Figure 5c (line 344) should be in bold. 
    • We thank the reviewer for identifying this formatting error and the change has been corrected.
  • In Figure 2b there is no titles on OY axis in 4 figures in the line. So, please, remove also titles on OY axis in figures in line with same titles in Figure 1b and Figure 2c.
    • We thank the reviewer for their comment to increase the consistency between figures and the requested change has been made.

Reviewer 3 Report

Comments and Suggestions for Authors

Occhiuto and Liby used the mouse Lewis lung carcinoma cell culture to assess the role of KEAP-1-NRF2 signaling axis in the regulation of anti-tumor immunity. For that purpose, they produced several clones of Lewis carcinoma cell clones with the knockout of KEAP-1 gene. Besides expected upregulation of NRF2 protein levels, these clones exhibited the increase of the expression of several myeloid recruiting cytokines and immunosuppressive cell surface markers. In the mouse orthotopic model of lung cancer, KEAP-1 KO tumors demonstrated the decreased infiltration of T-lymphocytes and increased infiltration of M2 macrophages. The RNAseq analysis indicates a possibility of the increase in KEAP-1 KO cells of the prostaglandin E2, which is known to suppress anti-tumor immunity. This is a solid study, with interesting novel results.

Critiques:

1.     The paper does not contain the information about the speed of tumor growth and formation of metastases.

2.     Prostaglandin related data are indirect. The authors should use a commercially available PGE2 ELISA kit to compare the production and secretion of PGE2 in KEAP-1 KO and WT cell cultures.

Author Response

  • The paper does not contain the information about the speed of tumor growth and formation of metastases.
    • This is an important aspect that we were unable to address using our current model. The LL2 cell line is a very aggressive cancer and forms tumors quickly in vivo. Given their rapid development, we were unable to assess differences in tumor growth with such fast kinetics; the total duration only spans 21 days. This, in combination with the inaccessibility of tumors within the pleural cavity, made assessment of tumor growth over time unfeasible. While unable to assess tumor growth, our model did allow us to evaluate the immune microenvironment in the orthotopic lung, which was one of the main goals of this study. However, it has been established in the literature that cancers with KEAP1 mutations have increased tumor growth (Zavitsanou et al.) and increased metastasis (Nrf2 Activation Promotes Lung Cancer Metastasis by Inhibiting the Degradation of Bach1, Lignitto et al. 2019; PMID: 31257023). Finally, we believe that metastasis is critically important in cancer progression but is outside of the scope of this current study which sought to evaluate the immune microenvironment of the primary tumor. It is a current future direction for our work.
  • Prostaglandin related data are indirect. The authors should use a commercially available PGE2 ELISA kit to compare the production and secretion of PGE2 in KEAP-1 KO and WT cell cultures.
    • We agree with the reviewer that the prostaglandin data is indirect. However, while we know that this pathway is altered in the cancer cells and macrophages in cancer models, we are exploring the specific changes in future studies. We believe that the signaling of prostaglandins and lipid mediators are important in KEAP1-mutant cancers. While our hypothesis is that PGE2 is mediating these effects, we believe further investigation is needed to confirm this. In place of a prostaglandin ELISA, we have added a western blot measuring expression of the COX-2 enzyme, which is the rate limiting step in prostaglandin synthesis. It has been reported that COX-2 expression highly correlates with PGE2 release and COX-2 overexpression is a poor prognostic indicator in lung cancers. We have added a paragraph at the end of the discussion regarding our data and its relevance.

Round 2

Reviewer 3 Report

Comments and Suggestions for Authors

The authors properly addressed the critiques